# Oxostephanine, Thalmiculine, and Thaliphyline—Three Isoquinoleine Alkaloids That Inhibit L-Type Voltage-Gated Ca²⁺ Channels

Jacinthe Frangieh [1,2,†], Claire Legendre [1,†], Dimitri Bréard [3], Pascal Richomme [3], Daniel Henrion [1], Ziad Fajloun [2], César Mattei [1], Anne-Marie Le Ray [3] and Christian Legros [3,*]

1 University of Angers, INSERM, CNRS, MITOVASC, Equipe CarME, SFR ICAT, F-49000 Angers, France; jacynthefrangieh@gmail.com (J.F.); claire.legendre@univ-angers.fr (C.L.); daniel.henrion@univ-angers.fr (D.H.); cesar.mattei@univ-angers.fr (C.M.)
2 Laboratory of Applied Biotechnology, AZM Centre for Research in Biotechnology and its Application, Doctoral School for Sciences and Technology, Tripoli 1300, Lebanon; zfajloun@gmail.com
3 University of Angers, SONAS, SFR QUASAV, F-49000 Angers, France; dimitri.breard@univ-angers.fr (D.B.); pascal.richomme@univ-angers.fr (P.R.); anne-marie.leray@univ-angers.fr (A.-M.L.R.)
* Correspondence: christian.legros@univ-angers.fr
† These authors contributed equally to this work.

**Abstract:** The isoquinoline alkaloids (IAs) represent a large and diverse subfamily of phytochemicals in terms of structures and pharmacological activities, including ion channel inhibition. Several IAs, such as liriodenine (an oxoaporphine) and curine (a bisbenzylisoquinoline (BBIQ), inhibit the L-type voltage-gated Ca²⁺ channels (LTCC). In this study, we aimed to search for new blockers of LTCC, which are therapeutic targets in neurological and cardiovascular diseases. We set up a screening assay using the rat pituitary GH3b6 cell line, which expresses two LTCC isoforms, Ca$_V$1.2 and Ca$_V$1.3. Both LTCC subtypes can be indirectly activated by KCl concentration elevation or directly by the dihydropyridine (DHP), BAY K8644, leading to an increase in the intracellular Ca²⁺ concentration ([Ca²⁺]$_i$). These Ca²⁺ responses were completely blocked by the selective LTCC DHP inhibitor, nifedipine. Thereby, 16 selected IAs were tested for their ability to inhibit KCl and BAY K8644-induced Ca²⁺ responses. We then identified three new potent LTCC blockers, namely, oxostephanine, thaliphyline, and thalmiculine. They inhibited LTCC with IC$_{50}$ values in the micromolar range through interaction to a binding site different to that of dihydropyridines. The two subfamilies of IAs, oxoaporphine with oxostephanine, and BBIQs with both thalyphilline and thalmiculine, constitute interesting pharmacophores for the development of future therapeutic leads for neurological and cardiovascular diseases.

**Keywords:** isoquinoline alkaloids; L-type voltage-gated Ca²⁺ channels; intracellular Ca²⁺ measurement; Fura-2 fluorescent probe; GH3b6 cells

## 1. Introduction

Voltage-gated Ca²⁺ channels (Ca$_V$ channels) control Ca²⁺ influx through the plasma membrane in response to depolarization and thereby contribute to the regulation of the intracellular Ca²⁺ concentration ([Ca²⁺]$_i$) in excitable cells [1,2]. Ca$_V$ channels are key players in Ca²⁺ signaling and allow physiological couplings, such as excitation secretion and neurotransmitter release, contraction, and transcription in endocrine cells, neurons, and myocytes. Ca$_V$ channels are heteromultimeric membrane proteins, consisting of a large principal α1 subunit, which contains the pore-forming domain, the voltage sensor, and gating apparatus, associated with the auxiliary subunits: α2δ, β, and γ [1,3]. The α1 subunit is encoded by 10 genes, namely, *cacna1s* (Ca$_V$1.1), *cacna1c* (Ca$_V$1.2), *cacna1d* (Ca$_V$1.3), *cacna1f* (Ca$_V$1.4), *cacna1a* (Ca$_V$2.1), *cacna1b* (Ca$_V$2.2), *cacna1e* (Ca$_V$2.3), *cacna1g* (Ca$_V$3.1), *cacna1h* (Ca$_V$3.2), and *cacna1i* (Ca$_V$3.3) [1,3].

Because of their involvement in various physiological and pathophysiological processes, the pharmacology of the $Ca_V1$ channel family, also called high-voltage and long-lasting or L-type $Ca_V$ channels (LTCC), including $Ca_V1.1$-4, has attracted attention for decades [1,3]. These four LTCC isoforms exhibit a tissue-specific expression. The $Ca_V1.1$ channel subtype is specifically expressed in skeletal myocytes and then involved in muscle contraction. The $Ca_V1.2$ and $Ca_V1.3$ channel subtypes exhibit a broader expression pattern; they are expressed in cardiomyocytes, smooth muscle cells (SMCs), neurons, and endocrine cells. These two LTCC isoforms contribute to various functions, such as memory, neurotransmitters release, myogenic tone, pace-making activity, and hormone secretion. The last LTCC isoform, $Ca_V1.4$, is specifically expressed in the retina and involved in the neurotransmitter release [1,3]. Variants of LTCC genes and or alteration of their expression have been correlated with severe muscular, cardiovascular, and neuronal channelopathies, such as hypokaliemic periodic paralysis, Timothy syndrome, cardiac arrythmia, deafness, arterial hypertension, Parkinson's disease, and neuropsychiatric disorders [1].

Three chemical classes of LTCC blockers, including dihydropyridine (DHP: e.g., nifedipine), phenylalkylamines (e.g., verapamil), and benzothiazepines (e.g., diltiazem) are extensively used for the treatment of cardiovascular diseases (CVDs), such as arrythmias, angina pectoris or myocardial ischemia, and arterial hypertension [1,4]. The 1,4-DHP reduces the contractility of vascular smooth muscle cells, resulting in vasodilation and thus efficiently decreases blood pressure [1,4]. These effects are mediated by the inhibition of $Ca_V1.2$ channels. The other beneficial effect of DHP in the treatment of arterial hypertension in aldosteronism is mediated by the inhibition of Cav1.3 channels of zona glomerula cells of the adrenal cortex, leading to the decrease in aldosterone secretion [5]. While the activators of LTCC have no therapeutical applications, they constitute useful tools for research studies. BAY K8644 (1,4-dihydro-2,6-dimethyl-5-nitro-4-[2-(trifluoromethyl)phenyl]-3-pyridinecarboxylic acid methyl ester) is one of the most used DHP to efficiently activate $Ca_V1.2$, $Ca_V1.3$, and $Ca_V1.4$ channels [1,4].

Various alkaloids behave as LTCC blockers and have been proven beneficial for the treatment of arterial hypertension [6,7]. Most of them belong to the large group of isoquinoline alkaloids (IAs) [8]. For example, tetrandine, a bisbenzylisoquinoline (BBIQ) alkaloid from *Stephania tetranda*, has been used in China since the 1950s for the treatment of angina pectoris and arterial hypertension [6,7,9,10]. Tetrandrine exhibits interesting pharmacological properties through the interaction with LTCC and other molecular targets, such as T-type $Ca_V$ channels, $Ca^{2+}$-activated $K^+$ channels, and muscarinic receptors [11]. Others BBIQs including curine, hernandezine, cepharanthine, daphnoline, and daurisoline have been reported to inhibit LTCC [12–15]. Curine induces an LTCC blockade and subsequent arterial vasodilation [14,15]. IAs from other subgroups, such as liriodenine (an oxoaporphine) and berberine and coptisine (two protoberberines), have been also shown to block LTCC [16–18].

Here, we aimed to characterize the ability of IAs from an in-house alkaloid library, for their ability to block LTCC. We selected 16 IAs from the BBIQ, oxoaporphine, berberine, protoberberine, and protopine subgroups, which display members already reported as inhibitors of LTCC. Two cularines and one pavine were also used, but no LTCC inhibition has been reported for these two IA subgroups. In our cell assay, we used the clonal cell line GH3b6, which endogenously expressed two LTCC isoforms, $Ca_V1.2$ and $Ca_V1.3$ [19,20]. We used KCl and (±)-BAY K8644 (BAY K8644) to activate LTCC, which both increased $[Ca^{2+}]_i$ in the GH3b6 cells, monitored by Fura-2 fluorescence. We determined that one oxoaporphine, oxostephanine, and two BBIQs, thalmiculine and thaliphyline, potently inhibit LTCC with different modes of action.

## 2. Materials and Methods

### 2.1. Reagents

Fura-2/AM probe and Pluronic®-F127 were purchased from Invitrogen (Carlsbad, CA, USA). Nifedipine, (±)-BAY K8644 (1,4-Dihydro-2,6-dimethyl-5-nitro-4-(2 trifluoromethylphenyl)

pyridine-3-carboxylic acid methyl ester), and all solvents were purchased from Sigma-Aldrich (Saint-Louis, MO, USA) or Fisher Scientific (Waltham, MA, USA).

### 2.2. Alkaloids Tested for Their Ability to Block LTCC

Sixteen alkaloid members of the BBIQ, oxoaporphine, berberine, protoberberine, protopine, pavine and cularine subgroups were selected in our in-house library (Table 1). These alkaloids were extracted and purified from plants and collected during the recent decades in the SONAS laboratory (EA 921, University of Angers, Angers, France). Each sample was solubilized in DMSO at 10 mM and stored at −20 °C in aliquots to prevent thaw/refreeze cycle for further usage.

### 2.3. Cell Culture

GH3b6 cell line is a subclone of a cell line isolated from a tumor of the rat pituitary gland [21]. This cell line was generously provided by Dr. Françoise Macari (IGF, Montpellier, France). Cells were cultured as previously described [20]. In brief, culturing was carried out from passage 14 until passage 32 at 37 °C with 5% $CO_2$ and 95% humidity in Dubelcco's Modified Eagle Medium/F12 (Lonza, Basel, Switzerland) supplemented with 10% fetal bovine serum (Lonza), 1 mM L-glutamine, and 1% of penicillin/streptomycin. GH3b6 cells were seeded at a density of 100,000 cells per well on a 96-well Greiner black/clear bottom imaging plates (Corning Hazebrouck, Borre, France) and incubated at 37 °C/5% $CO_2$ for 24 h before starting experiment.

**Table 1.** Isoquinoline alkaloids (IAs) used in this work to screen for LTCC blockers.

| Trivial Name | Chemical Class | Structure | Plant Origin (Part of Plant) |
|---|---|---|---|
| (+)-cepharanthine [22] | BBIQ |  | *Stephania suberosa, Menispermaceae* (tuberous roots) |
| thaliphylline [23] | BBIQ |  | *Thalictrum minus var. microphyllum, Ranunculaceae* (roots and rhizomes) |
| (−)-curine [24] | BBIQ |  | *Cyclea barbata, Menispermaceae* (roots) |
| thalmirabine [24] | BBIQ |  | *Thalictrum minus var. microphyllum, Ranunculaceae* (roots and rhizomes) |
| (−)-thalmiculine [25] | BBIQ |  | *Thalictrum cultratum, Ranunculaceae* (whole plant) |
| (+)-bebeerine [26] | BBIQ |  | *Curarea candicans, Menispermaceae* (roots) |
| (−)-curicycleatjenine [27] | Amidic BBIQ |  | *Cyclea atjehensis, Menispermaceae* (whole plant) |

**Table 1.** *Cont.*

| | | | |
|---|---|---|---|
| (+/−)-8-oxotetrahydropalmatine * | Protoberberine |  | *Pycnarrhena* sp., *Menispermaceae* (unknown) |
| tTetrahydropalmatine * | Protoberberine |  | *Berberis* sp., *Berberidaceae* (unknown) |
| protopine [28] | Protopine |  | *Corydalis majori, Fumariaceae* (whole plant) |
| oxostephanine [29] | Oxoaporphine |  | *Stephania venosa, Menispermaceae* (leaves) |
| liriodenine [29] | Oxoaporphine |  | *Stephania venosa, Menispermaceae* (leaves) |
| (+)-claviculine [28] | Cularine |  | *Corydalis claviculata, Papaveraceae* (whole plant) |
| (+)-celtine [30] | Cularine |  | *Ceratocapnos palaestinus, Fumariaceae* (whole plant) |
| (−)-norargemonine [27] | Pavine |  | *Cyclea atjehensis, Menispermaceae* (whole plant) |
| (S)-stylopine [31] | Berberine |  | *Corydalis majori, Fumariaceae* (whole plant) |

* unpublished data.

## 2.4. Quantitative Real-Time PCR

Total RNA from GH3b6 cells was extracted from using the RNeasy micro kit (Qiagen, Courtaboeuf, France). One μg of total RNA was processed for cDNA synthesis using random hexamers and the QuantiTect Reverse Transcription kit (Qiagen). Real-time PCR assays were assessed on a LightCycler 480 Instrument II (Roche, Meylan, France) using Sybr® Select Master Mix (Applied Biosystems®, Waltham, MA, USA) and 10 ng of cDNA and gene-specific primers of LTCC (Table 2). Relative quantification of gene expression was normalized to the mean of expression of two housekeeping genes *gapdh* (glyceraldehyde-3-phosphate dehydrogenase) and *gusb* (beta-glucuronidase) (Table 2) using the $2^{-\Delta Ct}$ method, where Ct is the threshold cycle, as previously described [20].

**Table 2.** Primer pairs used to amplify LTCC transcripts by RTq-PCR.

| Gene Name | GenBank Accession Number | Protein Name | Forward Primer (5′-3′) | Reverse Primer (5′-3′) |
|---|---|---|---|---|
| *cacna1s* | NM_05873 | Ca$_V$1.1 | gcgtcgtggagtggaaac | gggagctggagtttaagaatga |
| *cacna1c* | NM_012517 | Ca$_V$1.2 | tggctcacagaagtgcaaga | agcatttctgccgtgaaaag |
| *cacna1d* | NM_017298 | Ca$_V$1.3 | ccatgctcactgtgttccag | ctcccatcctatcgcatcat |
| *gusb* | NM_017015.2 | Gus | ctctggtggccttacctgat | cagactcaggtgttgtcatcg |
| *gapdh* | NM_017008.3 | Gapdh | tgggaagctggtcatcaac | gcatcaccccatttgatgtt |

Primer pairs used to amplify LTCC transcripts by RTq-PCR. GenBank accession number, sequence, and amplicon size are listed. The primer sequences for the housekeeping genes *gusb* and *gapdh* are also indicated.

### 2.5. Monitoring Intracellular Ca²⁺ Using Fura-2

After 24 h of incubation, cells were washed with HBSS (Hank's balanced salt solution) and then incubated with the dual excitation, monoemission, $Ca^{2+}$ probe, Fura-2 AM (4 µM), and Pluronic®-F127 (0.02%) in Fura-2 buffer solution composed of (in mM): 2.5 $CaCl_2$, 1 $MgCl_2$, 10 HEPES, and 0.5% BSA (Bovine serum albumin) (pH = 7.4) for 1 h at room temperature and in dark conditions. After washing twice with HBSS, cells were incubated with Fura-2 buffer solution for 1 h. Plates were then illuminated at 340 nm ($Ca^{2+}$-bound Fura-2) and 380 nm ($Ca^{2+}$-free Fura-2) excitation wavelengths, and the resulting fluorescence emission was monitored at 510 nm using a FlexStation® 3 Benchtop Multi-Mode Microplate Reader (Molecular Devices, Sunnyvale, CA, USA). Thirty seconds after baseline recording, LTCC activators, including KCl (1, 10, 30, and 100 mM) or BAY K8644 (1 µM), were automatically injected with or without nifedipine, and the fluorescence emission was read between 220 and 400 s. For testing the ability of IAs to inhibit LTCC, each IA was coinjected with KCl or BAY K8644 at 30 s, and the fluorescence of Fura-2 was monitored as described above. As negative control, HBSS was injected alone, as was nifedipine and each IA. Each condition was performed in triplicate wells and repeated at least twice. The membrane depolarization induced by 10, 30, and 100 mM was calculated with Goldman–Hodgkin–Katz equation:

$$E = \frac{RT}{F} Ln \left( \frac{P_K \left[ K^+ \right]_o + P_{Na} \left[ Na^+ \right]_o + P_{Cl} \left[ Cl^- \right]_o}{P_K \left[ K^+ \right]_i + P_{Na} \left[ Na^+ \right]_i + P_{Cl} \left[ Cl^- \right]_i} \right)$$

E is the membrane potential; $R$ is the universal gas constant; $T$ is the temperature in Kelvin; $F$ is the Faraday's constant; $P_K$, $P_{Na}$ and $P_{Cl}$ are the membrane permeability values for $K^+$, $Na^+$, and $Cl^-$, respectively; $o$: outside; and $i$: intracellular.

### 2.6. Statistical Analysis

Fluorescence data analysis was performed using the SoftMax Pro 5.4.1 software (Molecular Devices, Sunnyvale, CA, USA). The kinetic traces of $Ca^{2+}$ signals, monitored with Fura-2, were plotted as a ratio of fluorescence intensities measured at λ = 510 nm as emission, after a dual excitation at λ = 340 nm and 380 nm. The area under curve (AUC) was used for analysis. All graphs and statistical analysis were established using GraphPad Prism 7.02 (GraphPad Software, San Diego, CA, USA). Data are presented as mean ± SEM. Nonlinear analysis was used to fit the concentration–response data with Langmuir–Hill equation with Hill coefficient. For this analysis, the integration of the fluorescence kinetics (AUC) obtained with increasing concentrations of ligands were used. $IC_{50}$ or $EC_{50}$ and Hill coefficient values were calculated from at least two experiments. When applicable, nonparametric tests were used to analyze significance.

## 3. Results

### 3.1. KCl and BAY K8644 Induced Ca²⁺ Responses Mediated by L-Type Ca_V Channels (LTCC) in GH3b6 Cells

We first evaluated Cav channel expression by RT-qPCR (Figure 1). The data showed the endogenous expression of *cacna1c* and *cacna1d* transcripts, encoding for $Ca_V1.2$ and $Ca_V1.3$ channel subtypes, respectively (Figure 1). The mRNA expression level of *cacna1c* was found significantly higher than that of *cacna1d* ($p < 0.0001$, one-way ANOVA). As expected, the *cacna1s* mRNA encoding the muscular LTCC isoform $Ca_V1.1$ was not detected.

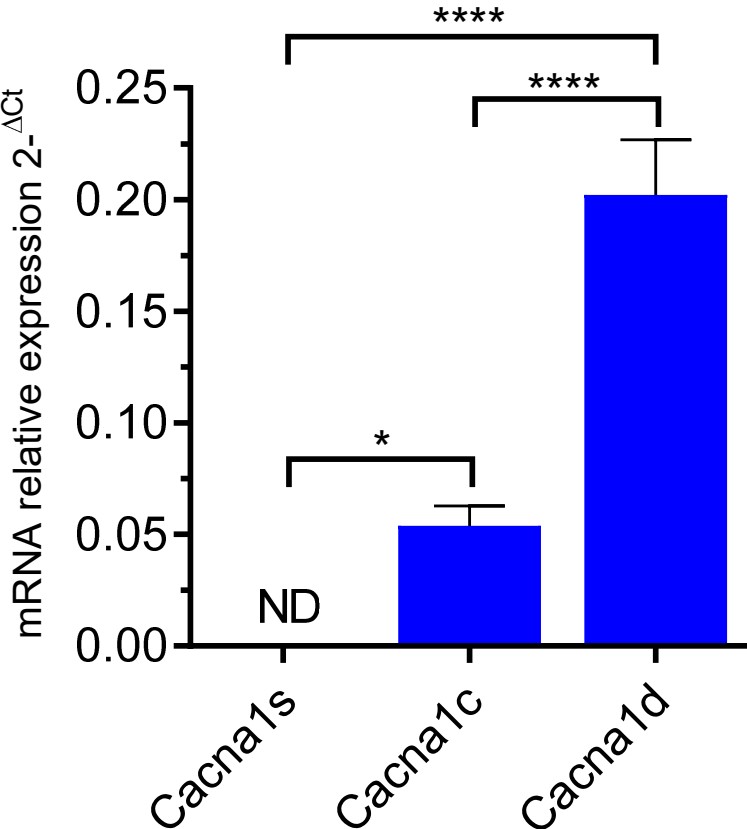

**Figure 1.** RT-qPCR amplification of LTCC genes in GH3b6 cells. The histogram illustrates the expression level of *cacna1s, cacna1c,* and *cacna1d* encoding $Ca_v1.1$, $Ca_v1.2$, and $Ca_v1.3$, respectively. The data are mean of mRNA relative expression ±SEM. ND: Not Detected. One-way ANOVA ($p < 0.0001$) followed by Tukey's post hoc multiple comparison test was performed. * $p < 0.05$, **** $p < 0.0001$.

Next, we measured the variation of $[Ca^{2+}]_i$ in responses to membrane depolarization with KCl (Figure 2a). The data were best fitted to a bell-shaped concentration–effect curve with a maximal response at 10 mM KCl (Figure 2b). Thus, we tested the ability of nifedipine to inhibit the $[Ca^{2+}]_i$ elevation induced by 10 mM KCl (Figure 2c). At 10 μM, nifedipine abolished this response (93.9 ± 1.7% inhibition, Figure 2d). This indicates that the $[Ca^{2+}]_i$ elevation induced by KCl was mostly mediated by LTCC.

We then used BAY K8644 to activate LTCC. As illustrated in Figure 3a, BAY K8644 induced $[Ca^{2+}]_i$ elevations in a concentration-dependent manner with an $EC_{50}$ of 1.07 ± 0.04 μM and a Hill coefficient of 1.18 ± 0.15 ($n = 2$, Figure 3a,b). The increase in $[Ca^{2+}]_i$ induced by BAY K8644 was inhibited by nifedipine in a concentration-dependent manner with an $IC_{50}$ of 1.69 ± 0.51 μM and a Hill coefficient of 1.94 ± 0.16, respectively ($n = 2$, Figure 3c,d). At 10 μM, nifedipine totally inhibited the increase in $[Ca^{2+}]_i$ induced by BAY K8644. These data demonstrate that the $[Ca^{2+}]_i$ elevation in response to BAY K8644 is fully mediated by LTCC.

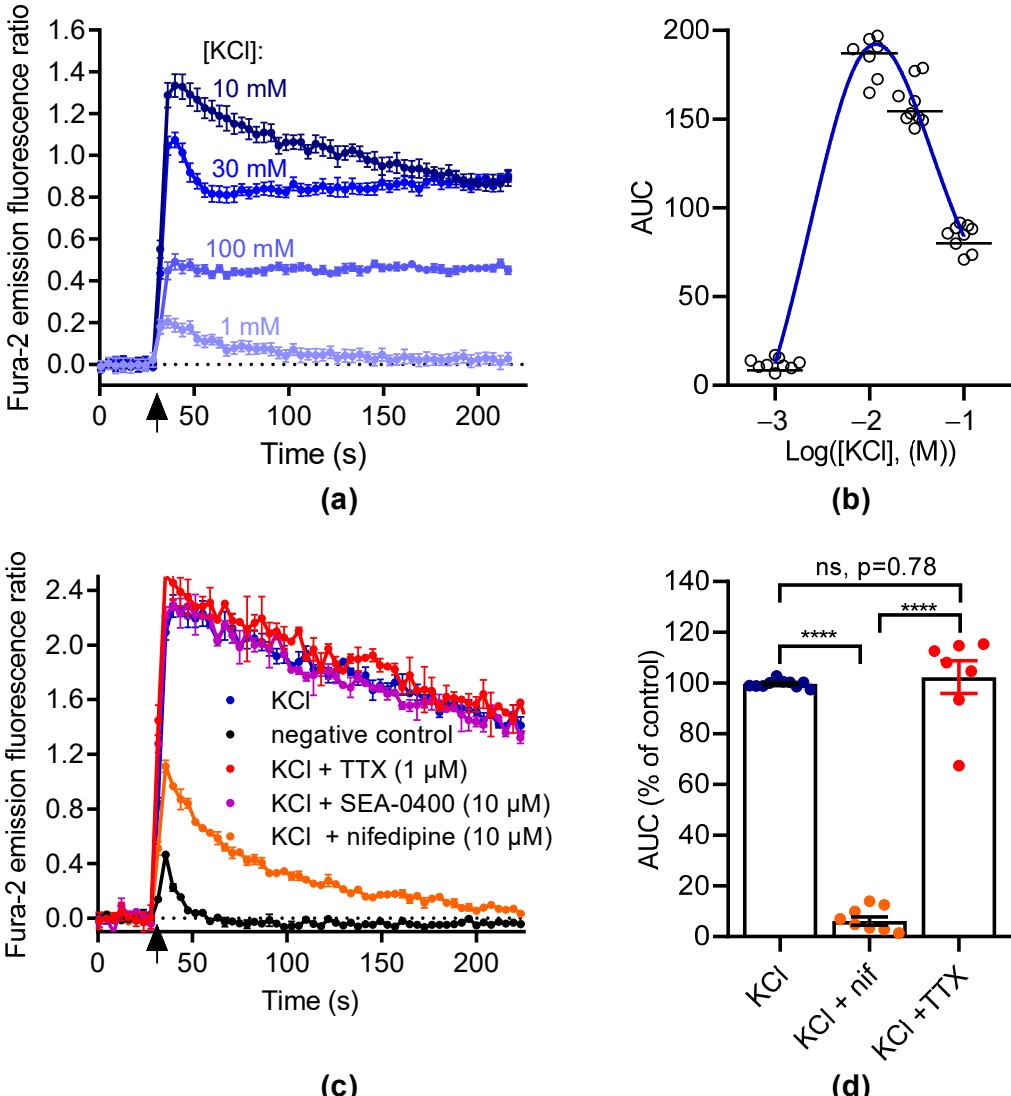

**Figure 2.** KCl induces intracellular Ca$^{2+}$ increase in GH3b6 cells. (**a**) Representative kinetic traces of KCl-induced Ca$^{2+}$ responses. The fluorescence emission ratio of Fura-2 increased after the injection of KCl (black arrow) at 1, 10, 30, and 100 mM, respectively. (**b**) The concentration–Ca$^{2+}$ response relationship for KCl is described by a bell-shaped curve. The AUC of kinetic traces of KCl-induced Ca$^{2+}$ responses were fitted by a bell-shaped curve equation. The horizontal dashes represent the mean value. (**c**) Representative kinetic traces of Ca$^{2+}$ responses induced by KCl at 10 mM in the absence and presence of nifedipine (10 μM), TTX (1 μM), and SEA-0400 (10 μM), respectively. As a negative control, HBSS was injected alone. (**d**) The scatter plot illustrates the effects of nifedipine (10 μM) and TTX (1 μM) on Ca$^{2+}$ responses induced by 10 mM KCl. The AUC of fluorescence emission traces over time were normalized using maximum response to 10 mM KCl. The bars represent the mean ± SEM. Significance between groups was analyzed by one-way ANOVA ($p < 0.0001$) followed by Tukey's post hoc multiple comparison test. ****, $p < 0.0001$; ns, nonsignificant. Data are shown as mean ± SEM ((**a**,**c**): $n = 3$; (**b**): $n = 9$).

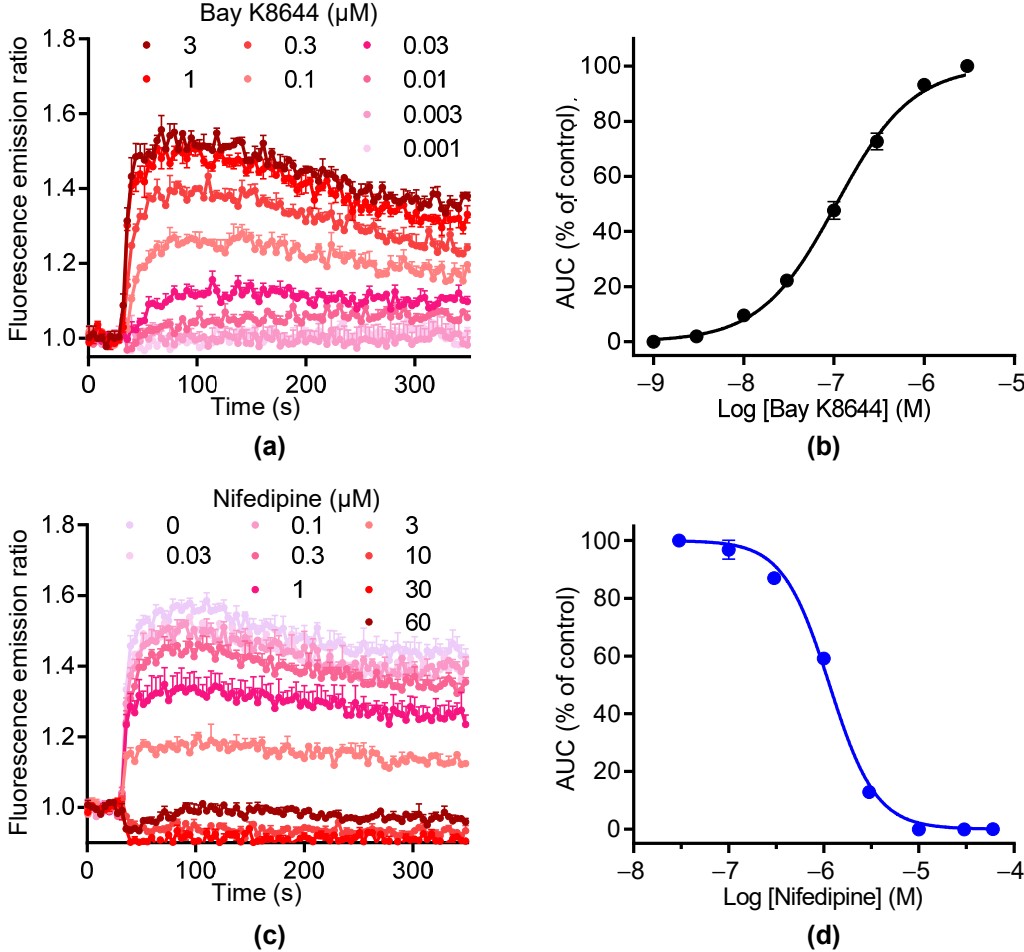

**Figure 3.** BAY K8644 activates LTCC channels in GH3b6 cells. (**a**) Example of kinetic traces of Fura-2 emission fluorescence induced by increasing concentrations of BAY K8644. (**b**) Concentration–response curve of BAY K8644-evoked $Ca^{2+}$ responses. (**c**) The kinetic traces show that the $Ca^{2+}$ responses induced by 1 μM of BAY K8644 were inhibited by nifedipine in a concentration-dependent manner. (**d**) Inhibition–concentration curve of nifedipine. (**b,d**) The AUC of kinetic traces were normalized to maximal response (% of control) and analyzed using Hill–Langmuir equation with variable slope. Data are means ± SEM (*n* = 3). These experiments were repeated twice.

### 3.2. Screening of Isoquinoline Alkaloids for Inhibitors of LTCC

Sixteen alkaloids (Table 1) were chosen to screen for LTTC inhibitors based on structural analogs known to inhibit LTCC. Each alkaloid was injected alone or coinjected at 10 μM with BAY K8644 or KCl. None of the 16 compounds injected alone modified the Fura-2 fluorescence emission ratio (Figure 4a). Oxostephanine, liriodenine, thaliphylline, and thalmiculine inhibited BAY K8644-induced $Ca^{2+}$ responses with higher potency, at 72.4 ± 5.9%, 44.0 ± 0.8%, 41.5 ± 8.9%, and 71.6 ± 11.0%, respectively (Figure 4b). With the exception of liriodenine, these IAs also inhibited KCl-induced $Ca^{2+}$ responses to the same extent (oxostephanine: 67.5 ± 8.8%, thaliphylline: 46.9 ± 3.8%, and thalmiculine: 60.0 ± 15.6%, Figure 4b). Liriodenine had a very low effect at 10 μM on KCl-induced $Ca^{2+}$ responses (4.9 ± 4.8%). Additionally, cepharanthine used at 10 μM had a strong inhibitory effect on KCl-induced $Ca^{2+}$ responses (59.5 ± 4.6%) and a low effect on BAY K8644-induced $Ca^{2+}$ responses (12.4 ± 1.8%).

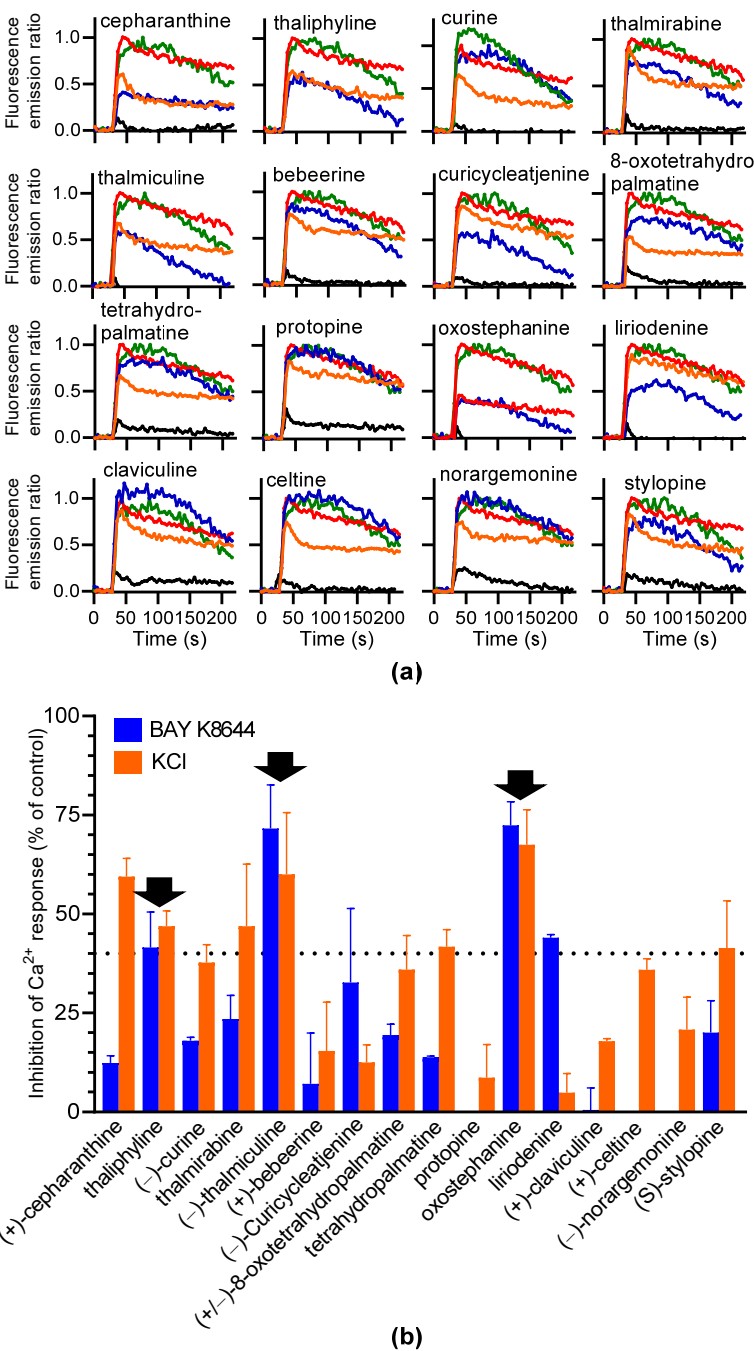

**Figure 4.** Screening of 16 IAs for inhibiting BAY K8644 or KCl-induced Ca$^{2+}$ response in GH3b6 cells. (**a**) The kinetic traces show the fluorescence emission ratio over time after injection of each IA (10 μM), alone (black line), with 1 μM of BAY K8644 (blue line), or with 10 mM of KCl (orange line). BAY K8644 (green line) and KCl (red line) were injected alone as positive controls. Data are means (*n* = 3). (**b**) The histogram shows the inhibitory effects in % of each IA on BAY K8644 and KCl-induced Ca$^{2+}$ responses. The black arrows point out the IAs, which inhibit both BAY K8644 and KCl -induced Ca$^{2+}$ responses at more than 40% (dashed horizontal line). Data are means ± SEM (*n* = 3–4).

We finally selected oxostephanine, thaliphylline, and thalmiculine, which inhibited both BAY K8644 and KCl-induced Ca$^{2+}$ responses at 40% minimum for further pharmacological characterization. The potency of the inhibitory effects of these three IAs on BAY K8644 (Figure 5a) or KCl-induced (Figure 5b) Ca$^{2+}$ increases was further characterized. The

determined IC$_{50}$ values were in the micromolar range for the three molecules and in the same range with both activators of LTCC (4–10 μM, Table 3).

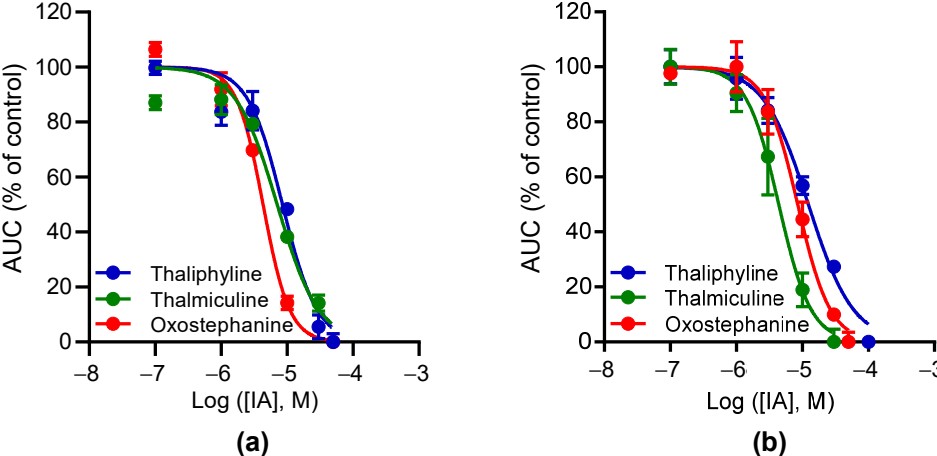

**Figure 5.** Concentration–inhibition curves for oxostephanine, thalmiculine, and thaliphyline. The curves represent the relationships of concentration–inhibition effects of oxostephanine, thalmiculine, and thaliphyline on Ca$^{2+}$ responses induced by 1 μM BAY K8644 (**a**) or by 10 mM KCl (**b**). Data (AUC of Ca$^{2+}$ kinetic response) were normalized to maximal response induced by BAY K8644 or by KCl (% of control) alone. Data were best fitted with Hill–Langmuir equation. Data were expressed as means ± SEM (*n* = 3). These experiments were repeated thrice.

**Table 3.** IC$_{50}$ and Hill slope values for oxostephanine, thaliphylline, and thalmiculine.

| IA Name | BAY K8644 | | KCl | |
| --- | --- | --- | --- | --- |
| | IC$_{50}$ (μM) | Hill Slope | IC$_{50}$ (μM) | Hill Slope |
| oxostephanine | 3.97 ± 0.54 | 1.87 ± 0.41 | 8.36 ± 0.86 | 1.80 ± 0.30 |
| thaliphylline | 9.21 ± 0.71 | 1.53 ± 0.69 | 10.97 ± 0.96 | 1.39 ± 0.51 |
| thalmiculine | 7.00 ± 1.10 | 2.00 ± 0.40 | 4.96 ± 0.98 | 1.55 ± 0.50 |

Fitting parameters of concentration–inhibition curves shown in Figure 5. Data are mean ± SEM of three independent experiments.

### 3.3. Effects of Oxostephanine, Thaliphylline, and Thalmiculine on the Concentration–Effect Relationship of BAY K8644

Next, we characterized the inhibitory effects of oxostephanine, thaliphylline and thalmiculine, at 10 μM each, on the concentration–effect relationship of BAY K8644 (Figure 6). Oxostephanine caused a right shift of the curve, leading to a 10-fold increase in the EC$_{50}$ of BAY K8644 (*p* < 0.05). In addition, the E$_{max}$ value of BAY K8644 was significantly decreased 2.6-fold in the presence of oxostephanine (*p* < 0.05, Mann–Whitney test). Thaliphylline and thalmiculine did not significantly change the EC$_{50}$ value of BAY K8644 (*p* = 0.9) (Table 4). However, these two BBIQs significantly decreased the E$_{max}$ value of BAY K8644 (3.7-fold, *p* < 0.05, with thalmiculine and 1.6-fold, <0.05, with thaliphyline, Mann–Whitney test). The E$_{max}$ value of BAY K8644 decreased to a smaller extent than observed for oxostephanine and thalmiculine (Table 4). These results show that oxostephanine and both BBIQs exhibit distinct mechanisms of inhibition; oxostephanine altered both potency and efficacy of BAY K8644, while thaliphylline and thalmiculine only decreased the efficacy of BAY K8644 (Figure 6).

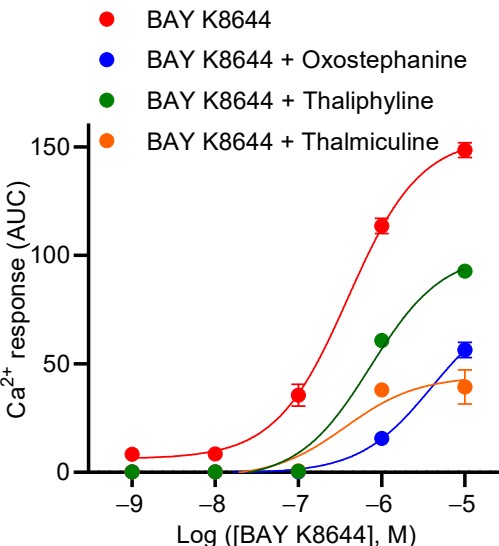

**Figure 6.** Concentration–response curves of BAY K8644 in the presence of oxostephanine, thalmiculine, or thaliphylline. A total of 10 μM of oxostephanine, thalmiculine, and thaliphylline were coinjected with increasing concentrations of BAY K8644. Data (AUC of $Ca^{2+}$ kinetic response) were best fitted with Hill–Langmuir equation and are shown in Table 4. Data are mean ± SEM ($n = 3$). These experiments were repeated thrice.

**Table 4.** Effects of oxostephanine, thaliphylline, and thalmiculine on potency and efficacy of BAY K8644.

| IA Name | $EC_{50}$ (μM) | $E_{max}$ (AUC) |
|---|---|---|
| Control | 0.40 ± 0.39 | 148.6 ± 0.05 |
| +10 μM oxostephanine | 4.00 ± 0.09 * | 56.45 * ± 0.10 |
| +10 μM thaliphylline | 0.73 ± 0.96 ns | 92.7 * ± 0.45 |
| +10 μM thalmiculine | 0.36 ± 0.10 ns | 39.33 * ± 0.94 |

Fitting parameters of concentration–effect curves of BAY K8644 shown in Figure 6. Mann–Whitney test was used to compare control and each IA. *, $p < 0.05$. Data are mean ± SEM of three independent experiments. ns: not significant.

*3.4. Inhibitory Effects of Oxostephanine, Thaliphylline, and Thalmiculine Increase versus KCl Concentration*

To further characterize the effects of oxostephanine, thaliphylline, and thalmiculine, each at 10 μM, on KCl-induced $Ca^{2+}$ responses, their inhibitory effects were evaluated as a function of the KCl concentrations (10, 30, and 100 mM) (Figure 7). According to the Goldman–Hodgkin–Katz equation, 10, 30, and 100 mM of KCl induce 8, 28, and 68 mV of depolarization. These experiments allowed us to establish the voltage-dependency of the inhibitory effects of oxostephanine, thaliphylline, and thalmiculine. For oxostephanine, the inhibition rate was similar on $Ca^{2+}$ responses induced by 10 mM (49.4 ± 3.5%), 30 mM (57.9 ± 3.9%), and 100 mM ($p = 0.24$, one-way ANOVA test). However, for thaliphylline, the inhibition rate was similar at 10 mM (52.7 ± 3.8%) and 30 mM KCl (48.2 ± 7.6%) ($p = 0.80$, one-way ANOVA test), but the inhibition rate was significantly higher at 100 mM (85.8 ± 0.4%), in comparison to 10 and 30 mM ($p < 0.001$, one-way ANOVA test). For thalmiculine, the inhibition rate was similar at 30 mM (78.2 ± 3.0%) and 100 mM KCl (89.7 ± 4.2%) ($p = 0.11$, one-way ANOVA test), but the inhibition rate was significantly lower at 10 mM (61.0 ± 2.4%) in comparison to 30 and 100 mM ($p < 0.05$ and $p < 0.001$, respectively, one-way ANOVA test). In conclusion, the inhibitory effects of thaliphylline and thalmiculine changed according the KCl concentrations, reflecting changes in membrane depolarization levels.

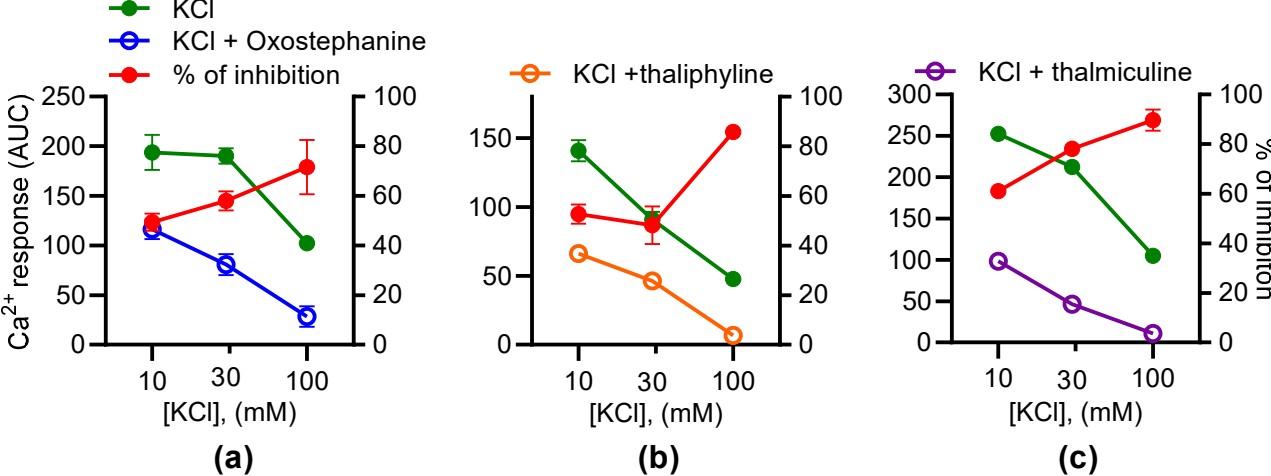

**Figure 7.** The inhibitory effects of oxostephanine, thaliphylline, and thalmiculine versus the KCl concentrations. The effects of 10 μM of oxostephanine (**a**), thaliphylline (**b**), and thalmiculine (**c**) were evaluated on $Ca^{2+}$ responses induced by increasing concentrations of KCl (10, 30, and 100 mM). Left y axis: AUC were determined from kinetics of the Fura-2 emission fluorescence ratio. Right axis: the inhibition was calculated as a ratio of AUC of kinetic traces in the presence and absence of IAs in %. Data are mean ± SEM (*n* = 3).

### 3.5. Structural Analogs of Oxostephanine, Thaliphylline, and Thalmiculine

Searching for structural analogs of oxostephanine allowed us to identify six other oxoaporphines alkaloids in the ChemIDplus data base (Figure 8a), namely, liriodenine, lauterine, atherospermidine, lanuginosine, oxocrebanine, and dicentrinone. These seven oxoaporphines only differ in the number and position of methoxy groups. The analysis of similarity using ChemMine tools [32] showed that oxostephanine shared the highest structural similarity with lauterine and atherospermidine, respectively (atom pair similarity of 0.83, Table 5). The analysis of these seven IAs by multidimensional scaling (MDS) (Figure 8a) highlighted at least three clusters: (1) oxostephanine and atherospermidine; (2) lanuginosine and lauterine; and (3) oxocrebanine and dicentrinone, while liriodenine appeared to be further than the other IAs.

**Table 5.** Structural similarities among oxostephanine and other oxoaporphines.

| Alkaloid Name | Oxosteph. | Liriodenine | Lauterine | Atherosperm. | Langinosine | Dicentrinone | Oxocreb. |
|---|---|---|---|---|---|---|---|
| oxosteph. | 1 | 0.73 | 0.83 | 0.83 | 0.75 | 0.66 | 0.75 |
| liriodenine | 0.73 | 1 | 0.70 | 0.72 | 0.71 | 0.53 | 0.49 |
| lauterine | 0.83 | 0.70 | 1 | 0.75 | 0.80 | 0.75 | 0.71 |
| atherosperm. | 0.83 | 0.75 | 0.75 | 1 | 0.70 | 0.62 | 0.68 |
| langinosine | 0.75 | 0.71 | 0.80 | 0.70 | 1 | 0.75 | 0.71 |
| dicentrinone | 0.66 | 0.53 | 0.75 | 0.62 | 0.75 | 1 | 0.85 |
| oxocreb. | 0.75 | 0.49 | 0.71 | 0.68 | 0.71 | 0.85 | 1 |

Atom pair (AP) similarities were computed using the online software https://chemminetools.ucr.edu/similarity/ (accessed on 2 May 2022) with the Tanimoto coefficient. oxosteph., oxostephanine; atherosperm., atherospermidine; oxocreb., oxocrebanine.

Since thaliphylline and thalmiculine belong to the BBIQ subgroup of IAs, we analyzed their structural similarities with tetrandrine and cepharanthine, which have been already described as LTCC blockers. While thaliphylline shared the highest structural similarity with tetrandrine (0.80), thalmiculine is structurally closer to tetrandrine (0.76) (Table 6). The MDS analysis did not reveal clustering of thaliphylline and thalmiculine but showed a relative high similarity between tetrandrine and cepharanthine (Figure 8b).

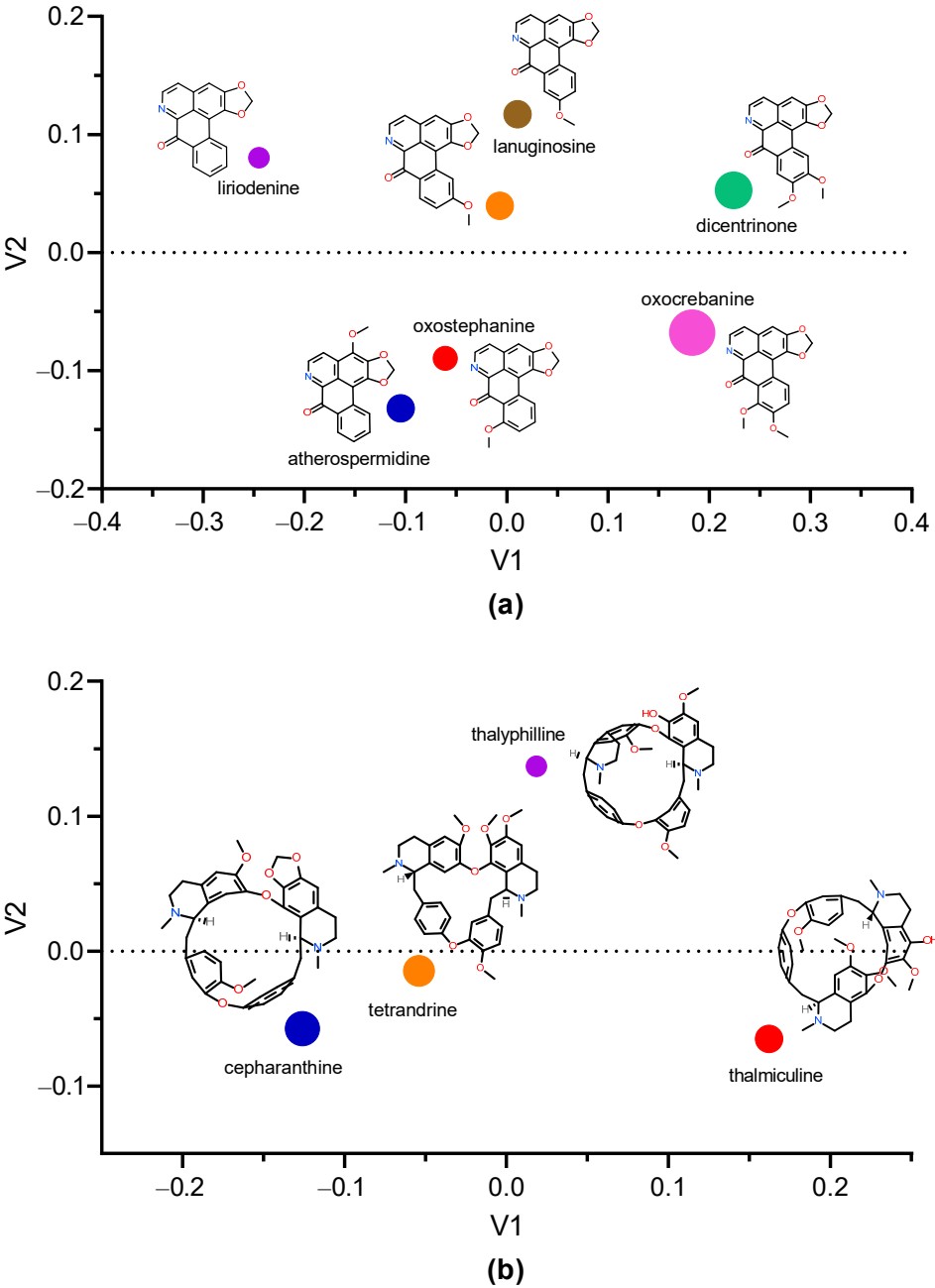

**Figure 8.** Multidimension scaling analysis using Tanimoto coefficient. The multidimension scaling analysis of structural similarity among selected IAs was performed with ChemMine tools [32]. (**a**) Clustering of oxostepharnine, liriodenine, lauterine, atherospermidine, lanuginosine, oxocrebanine, and dicentrinone. (**b**) Comparison of structures of thaliphylline and thalmiculine and the inhibitory effects of oxostephanine, thaliphylline, thalmiculine, tetrandrine, and cepharanthine versus KCl concentrations. Inside the graphs, the 2D structure of each IA is shown next to the corresponding colored dot, whose size is proportional to the molecular weight. V1 and V2 values were calculated using Tanimoto coefficient.

**Table 6.** Structural similarities among thaliphylline, thalmiculine, tetrandrine, and cepharanthine.

| Alkaloid Name | Thaliphylline | Thalmiculine | Tetrandrine | Cepharanthine |
|---------------|---------------|--------------|-------------|---------------|
| thaliphylline | 1 | 0.75 | 0.80 | 0.76 |
| thalmiculine | 0.75 | 1 | 0.76 | 0.71 |
| tetrandrine | 0.80 | 0.76 | 1 | 0.84 |
| cepharanthine | 0.76 | 0.71 | 0.84 | 1 |

Atom pair (AP) similarities were computed using the online software https://chemminetools.ucr.edu/similarity/ (accessed on 2 May 2022) with the Tanimoto coefficient.

## 4. Discussion

In this study, we identified three novel LTCC blockers, one oxoaporphine, oxostephanine, and two BBIQs, thalmiculine and thaliphyline, exhibiting structural analogy with known IAs, which inhibit LTCC. Oxostephanine, thaliphylline, and thalmiculine inhibited LTCC, expressed in GH3b6 cells with $IC_{50}$ values in the micromolar range. While oxostephanine decreased both potency and efficacy (maximal effect) in the activator of LTCC, BAY K8644, thalmiculine, and thaliphyline only altered its efficacy. In addition, the inhibitory effects of thalmiculine and thaliphyline increased as a function of the level of membrane depolarization induced by KCl, while oxostephanine did not. Altogether, these data revealed distinct modes of action inhibition of LTCC by both kinds of IAs.

GH3b6 cells endogenously express LTCC isoforms, $Ca_V 1.2$ and $Ca_V 1.3$, at the transcriptional level as previously described [19,20]. This was expected since this cell line is a subclone of GH3 cells, which have been extensively used for the investigation of LTCC pharmacology. Moreover, GH3b6 cells respond to both KCl and BAY K8644 by $[Ca^{2+}]_i$ elevations, which were suppressed by the selective LTCC blocker, nifedipine [1,4]. This constitutes strong pharmacological evidence that KCl and BAY K8644 only activate LTCC in GH3b6 cells, as it has previously been reported in GH3 cells, its clonal parental cells [33]. Thus, GH3b6 cells exhibit excitable properties and represent a convenient cellular model to screen for LTCC blockers by monitoring $[Ca^{2+}]_i$ using fluorescent $Ca^{2+}$ probes, such as Fura-2.

Screening assays based on fluorescent probes should be well-designed to avoid the selection of false positive hits and agonists or antagonists with weak potency. Data mining and analysis are easier and faster with fluorescent assays. That is particularly convenient in large primary screening for rapid selecting hits (Coquerel et al., https://www.mdpi.com/1420-3049/27/13/4133, accessed on 2 May 2022). Here, we benefited from a dual screening based on chemical and pharmacological activation of LTCC in an excitable cell line, allowing a rapid identification of relevant hits. The concept of $[Ca^{2+}]_i$ measurement using fluorescent probes for biomolecular screening has been extensively used but remains particularly powerful in ion channels targeting drug discovery [34,35]. Nevertheless, further experiments with electrophysiological techniques are unavoidable since key parameters, such as voltage-dependency and use-dependency, cannot be provided by fluorescent probe-based assays. Our approach consisting of the use of KCl to activate LTCC offers the advantage to induce membrane depolarization in a control manner.

Indeed, KCl-induced $Ca^{2+}$ responses in GH3 cells is well-known to depend on extracellular $Ca^{2+}$ and to be blockable by nifedipine [33]. However, extracellular $Na^+$ concentration could also influence KCl-induced $Ca^{2+}$ responses in these cells, since the replacement of extracellular $Na^+$ by NMDG or TRIS suppresses action potential in GH3 cells and $Ca^{2+}$ oscillations [36–38]. This is due to the contribution of the $Na^+$ leak background channel, NALCN, which regulates cell excitability as well as hormone secretion in endocrine cells [36]. Here, we found that high $K^+$ concentrations (30 and 100 mM) led to weaker $Ca^{2+}$ responses than low concentrations (10 mM). Since high $K^+$ concentrations were compensated by decreasing $Na^+$ concentrations, we believed that the decreases in KCl-induced $Ca^{2+}$ responses were likely due to low extracellular $Na^+$ concentrations and consecutively a decrease in NALCN depolarizing effects as we have previously demonstrated in this cell line [36].

Here, we found that the oxoaporphine, oxostephanine, represents a promising blocker of LTCC. Several pharmacological properties of oxostephanine [39] have already been described, such as antiproliferative effects against breast cancer cell lines [40] and antimicrobial [41,42] and antiviral agent activity [43]. However, ion channels have not yet been reported as targets for oxostephanine. Interestingly, its closest structural analog, liriodenine, without a methoxy group present in the benzyl group of oxostephanine, is well known to exhibit a plethora of biological activities mediated by binding to ion channels, including $Na_V$, $K_V$, $Ca_V$ channels and nicotinic and $GABA_A$ receptors [16,44,45]. Liriodenine had a weak inhibitory effect on KCl-induced $[Ca^{2+}]_i$ elevation in GH3b6 cells but was able to inhibit BAY K8644-induced $Ca^{2+}$ responses with a potency 2.5-fold weaker than oxostephanine (data not shown). These data agree with electrophysiological data, showing that liriodenine weakly inhibits (up to a maximum of 35%) the L-type $Ca^{2+}$ currents recorded in isolated cardiomyocytes [16]. Thus, oxostephanine could be considered as a stronger inhibitor of LTCC than its structural analog liriodenine.

It is conceivable that the high structural similarity between both oxoaporphines suggests common pharmacological targets, such as $Na_V$, $K_V$, $Ca_V$ channels and nicotinic and $GABA_A$ receptors. Other oxoaporphines have been described, sharing high structural similarities with oxostephanine, such as atherospermidine and lauterine. These three oxoaporphines only differ in the position of a methoxy group; oxostephanine and liriodenine exhibit similar pharmacological properties, such as anticancer and antibiotic activities, on different models [46–49]. Thus, we assume that the oxoaporphine skeleton represents an interesting pharmacophore targeting various ion channel types, and it would be very interesting to investigate further the structure–function relationships of their interaction with ion channels.

In contrast, no biological activities have been reported for the two BBIQs, thaliphylline and thalmiculine in the literature. Interestingly, thaliphylline and thalmiculine do not share high structural similarities, as the tetrandrine and cepharanthine pair. Thaliphylline appears to be structurally closer to tetrandrine than to thalmiculine. Thaliphylline and thalmiculine have been isolated from two different species of herbaceous perennial flowering plants in *Ranunculaceae*, *Thalictrum minus* var. *microphyllum* and *Thalictrum cultratum*, respectively, which contained various BBIQs [23,25]. Both BBIQs share 95% structural similarity with tetrandrine and cepharantine, which has been already described as a blocker of LTCC in GH3 cells with $IC_{50}$ in the micromolar range [12,50,51]. Tetrandrine binds to the benzothiazepine site of LTCC inhibitors, such as verapamil and diltiazem, and stimulates the binding of DHP LTCC inhibitors [12]. This is in accordance with our data showing that thaliphylline and thalmiculine bind to an allosteric site, distinct from that of DHP since both BBIQs strongly reduced the maximal effects of BAY K8644. Thus, our findings strongly suggest that thaliphylline and thalmiculine are two inhibitors of LTCC, similar to tetrandrine that is used in traditional Chinese folk medicine to treat hypertension and cardiovascular disease [11]. Then, it will be very interesting to determine whether thaliphylline and thalmiculine exhibit vasorelaxant effects like tetrandrine [9].

Oxostephanine induced a decrease in both potency and efficacy (maximal effect) of BAY K8644, reflecting a noncompetitive antagonism mechanism [52,53] and thus distinct binding sites for both molecules. In contrast, thaliphylline and thalmiculine altered only the efficacy (maximal effect) of BAY K8644. This is in agreement with the interaction of BBIQs with the benzothizepine site of LTCC [12] and previous data showing that benzothizepine and BAY K8644 bind with a noncompetitive mechanisms to LTCC [54]. Thus, we assumed that both BBIQs and BAY K8644 interact noncompetitively with LTCC. Further experiments will be necessary to established whether oxostephanine, thaliphylline, and thalmiculine are reversible or irreversible noncompetitive antagonists of BAY K8644 and also to decipher their binding sites.

Interestingly, the inhibitory effects of both BBIQs, thalyphilline and thalmiculine, on LTCC increased with KCl concentration, suggesting that their blocking mechanism depends on membrane depolarization. Since KCl-induced $Ca^{2+}$ responses were completely block by

nifedipine, the membrane depolarization induced by increasing KCl concentration $[Ca^{2+}]_i$ elevation is mediated by LTCC in GH3b6 cells. This is also a strong argument to conclude that thalyphilline and thalmiculine efficiently block LTCC. However, based on $IC_{50}$ values on the inhibition of $Ca^{2+}$ responses induced by 10 mM KCl and the comparison of inhibitory effects versus KCl concentrations, thalmiculine appeared to be a more potent inhibitor than thalyphilline. Further experiments will allow us to characterize further their mode of action and selectivity toward all LTCC subtypes. As tetrandrine, thalmiculine represents a valuable tool with potential therapeutic application for the treatment of neurological and cardiovascular diseases.

## 5. Conclusions

The major findings of this study are the discovery of the first biological activity of two BBIQs, namely, thaliphylline and thalmiculine, from two distinct species of *Thalictrum*, as novel blocker of LTCC. In addition, we also determined that oxostephanine, an oxoaporphine and structurally close to liriodenine, also exhibits potent inhibitory effects against LTCC. These three IAs likely bind to distinct DHP allosteric sites in LTCC. These data also highlighted the interest in screening two large subgroups of IAs, BBIQs, and oxoaporphines, for finding ligands of LTCC, which could be helpful in the development of novel drugs for the treatment of neurological and cardiovascular diseases.

**Author Contributions:** Conceptualization, C.L. (Christian Legros) and C.L. (Claire Legendre); methodology, C.L. (Christian Legros) and C.L. (Claire Legendre); formal analysis, C.L. (Christian Legros), C.L. (Claire Legendre) and J.F.; investigation, J.F., C.L. (Claire Legendre) and D.B.; writing—original draft preparation, C.L. (Christian Legros), C.L. (Claire Legendre) and J.F.; writing—review and editing, C.L. (Christian Legros), C.L. (Claire Legendre), C.M., Z.F., D.H., P.R., A.-M.L.R. and J.F.; supervision, C.L. (Christian Legros); project administration, C.L. (Christian Legros). All authors have read and agreed to the published version of the manuscript.

**Funding:** This research received no external funding.

**Institutional Review Board Statement:** Not applicable.

**Informed Consent Statement:** Not applicable.

**Data Availability Statement:** Not applicable.

**Acknowledgments:** We are grateful to Françoise Macari (IGF, Montpellier, France) for the GH3b6 cell line. We are also grateful to Linda Grimaud and Louis Gourdin for their technical assistance.

**Conflicts of Interest:** The authors declare no conflict of interest. The funders had no role in the design of the study; in the collection, analyses, or interpretation of data; in the writing of the manuscript; or in the decision to publish the results.

## Abbreviations

The following abbreviations are used in this manuscript:

| | |
|---|---|
| AUC | area under curve |
| BBIQ | bisbenzylisoquinoline |
| $Ca_V$ channels | voltage-gated $Ca^{2+}$ channels |
| IA | Isoquinoline Alkaloid |
| Cav channel | voltage-gated $Ca^{2+}$ channel |
| $[Ca^{2+}]_i$ | intracellular $Ca^{2+}$ concentration |
| gapdh | glyceraldehyde-3-phosphate dehydrogenase |
| gusb | beta-glucuronidase; |
| LTCC | L-type voltage-gated $Ca^{2+}$ channel |

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
