# Peer review of "Oxostephanine, Thalmiculine, and Thaliphyline—Three Isoquinoleine Alkaloids That Inhibit L-Type Voltage-Gated Ca2+ Channels"

_futurepharmacol, doi:10.3390/futurepharmacol2030016_

Round 1
Reviewer 1 Report
Manuscript review. ID: futurepharmacol-1733250, Oxostephanine, thalmiculine and thaliphyline, three isoquinoline alkaloids which inhibit L-type voltage-gated Ca2+ channels. By Frangieh Jacinthe , Legendre Claire , Bréard Dimitri , Richomme Pascal , Henrion Daniel , Fajloun Ziad , César Mattei , Le Ray Anne-Marie , Legros Christian
Authors describe the pharmacological characterization of a family of isoquinoleine alkaloids related to their ability to block voltaje-dependant Ca channels, specifically those classified as L-subtypes.
Overall, this is a nice classical pharmacological work that deserves to be published in a medium-rated journal. I could think that several different techniques should be performed to better characterize theses natural products as L-VGCC blockers, such as patch-clamp, radioligand binding assays, and so on. However, this manuscript has several merits, such as the experimental design, the clarity and conciseness. It can serve for young researchers as an introduction to the pharmacology of VGCC. Hence, authors were right to choose future pharmacology for submitting this manuscript.
For this reason, I recommend this manuscript for publication after a minor revision, where authors should respond some questions and issues I express.
The fluorescence-based Ca measurements is a widely accepted tool for testing new VGCC blockers. Nevertheless, authors should comment that this is an indirect assay. For instance, what if some of these blockers were indeed a sodium channel blockers?. They would probably exert a similar behavior, as they are affecting the depolarizing stimuli exacerbated by high K. I could agree that experiments with Bay K would respond this issue. But, what if these compounds could have an intracellular effect, for instance blocking the mitochondrial Na/Ca exchanger? Authors should state their opinion about these questions. In any case, I have to recognize that authors perform a very accurate controls, such as those plotted in figure 3, what reinforce the quality of the manuscript.
According to the protocol described in section 2.4, there is an interval of 1 h where the compounds are not present, that when authors washout the dye and subject cells just to the buffer solution for 1 h, without compounds. After that, compounds are co-injected with the depolarizing stimuli. This way seems to me very weird, since we could lose the previously incubated compounds, and we are only observed changes related to the co-incubation of cells.
As far a experiments of Figure 2a, authors should comment why 30 and 100 mM of K elicit a minor elevation of Ca signal than 10 mM.
Figure 1c, why does the injection of HBSS alone produce a tiny elevation of Ca around the second 40?
Minor points:
- The real VGCC agonist is the levo enantiomer of Bay K, i.e,. (S)-(-)-BAY K8644. The dextro enantiomer is indeed a VGCC blocker. For economic issues, it is widely sold as the racemic mixture because the agonist effect of the S-enantiomer is more potent than the blocking effect of the R-enantiomer.
- chapter 2.3.: Are authors sure that concentration of both antibiotics is 1 mM?. It seems too much.
- TYPOS:
- Line 37: regualtion
- Line 187: depolarizatoin
Author Response
Response to Reviewer 1
Dear Reviewer 1,
We thank you for your peer-reviewing, which helped us to improve this manuscript.
"Authors describe the pharmacological characterization of a family of isoquinoleine alkaloids related to their ability to block voltaje-dependant Ca channels, specifically those classified as L-subtypes.
Overall, this is a nice classical pharmacological work that deserves to be published in a medium-rated journal. I could think that several different techniques should be performed to better characterize theses natural products as L-VGCC blockers, such as patch-clamp, radioligand binding assays, and so on. However, this manuscript has several merits, such as the experimental design, the clarity and conciseness. It can serve for young researchers as an introduction to the pharmacology of VGCC. Hence, authors were right to choose future pharmacology for submitting this manuscript.
For this reason, I recommend this manuscript for publication after a minor revision, where authors should respond some questions and issues I express.
The fluorescence-based Ca measurements is a widely accepted tool for testing new VGCC blockers. Nevertheless, authors should comment that this is an indirect assay. For instance, what if some of these blockers were indeed a sodium channel blockers?. They would probably exert a similar behavior, as they are affecting the depolarizing stimuli exacerbated by high K.
I could agree that experiments with Bay K would respond this issue. But, what if these compounds could have an intracellular effect, for instance blocking the mitochondrial Na/Ca exchanger? Authors should state their opinion about these questions. In any case, I have to recognize that authors perform a very accurate controls, such as those plotted in figure 3, what reinforce the quality of the manuscript."
- Answer : thanks for these comments. We agree ! As said in the abstract (lines 21-23), we clearly stated that KCl stimulation indirectly activates LTCC, while Bay K8644 directly activate them. We have compiling evidence that KCl stimulation induced intracellular Ca2+ elevation through Ca2+ entry in GH3 cells. Vela et al (2006, doi:10.1152/ajpcell.00429.2006), have shown that in the absence of extracellular Ca2+, 12.5 mM of KCl does not induced intracellular Ca2+ elevation. Like us, they have shown that nifedipine inhibits KCl-induced Ca2+ responses, indicating that LTCC play a pivotal role in these responses to KCl. Concerning the role of mitochondria in Ca2+ signalling of endocrine cells (and other cell type too), we agree that they are certainly involved. We did not investigate this feature of Ca2+ homeostasis in our cell line, but we are taking into account that we could not exclude effects of the three IA reported in this work, if we consider only KCl-based assays. However, the fact that they abolished Bay K8644 Ca2+ response, indicated that they target LTCC.
We added in figure 2, experiments that we did in control to show that TTX, a Nav channel blocker s and also, that SEA-0400, a NCX blocker did not affect KCl-induced Ca2+ response. Concerning the effects of oxostephanine, liriodenine and thalmiculien, we have a manuscript in revision in Molecules (manuscript molecules-1590145), showing that they have no effects on Nav channels at 10 µM. We did not mention these data since they are not yet published.
However, we believe that the alkaloids could have several molecular targets and it would be very interesting to test them on mitochondrial NCX (mNCX) for example, with an appropriate assay. Del Viscovo et al. (2012, doi:10.1152/ajpendo.00389.2011) have proven that mNCX contributes to intracellular Ca2+ increase in response to iodotyronines in GH3. The mechanism underlying this might be distinct than that involving a direct membrane depolarization (iodotyronines activate NO and akt pathcways). However, it would be interesting to test the effects of mNCX inhibitors on KCl-induced Ca2+ response. We added a sentence in the discussion.
"According to the protocol described in section 2.4, there is an interval of 1 h where the compounds are not present, that when authors washout the dye and subject cells just to the buffer solution for 1 h, without compounds. After that, compounds are co-injected with the depolarizing stimuli. This way seems to me very weird, since we could lose the previously incubated compounds, and we are only observed changes related to the co-incubation of cells."
Answer : Indeed, we used the Ca2+ probe Fura-2 AM. This acetoxymethyl (AM) ester is cell permeant and is hydrolyzed to Fura-2 by endogenous esterases once it enters cells. Fura-2 could not cross membrane. Thus, washing does not lead to lose of Fura-2.
"As far a experiments of Figure 2a, authors should comment why 30 and 100 mM of K elicit a minor elevation of Ca signal than 10 mM."
Answer : We added a new paragraph to explain this effects of K+ concentration in the discussion:
“Indeed, KCl-induced Ca2+ responses in GH3 cells is well-known to depend on extracellular Ca2+ and to be blockable by nifedipine [33]. However, extracellular Na+ concentration could also influence KCl-induced Ca2+ responses in these cells, since replacement of extracellular Na+ by NMDG or TRIS suppresses action potential in GH3 cells and Ca2+ oscillations [36–38]. This is due to the contribution of the Na+ leak background channel, NALCN, which regulates cell excitability as well as hormone secretion in endocrine cells [36]. Here, we found that high K+ concentrations (30 and 100 mM) led to weaker Ca2+ responses than low concentration (10 mM). Since high K+ concentrations were compensated by decreasing Na+ concentrations, we believed that the decrease of KCl-induced Ca2+ responses were likely due to low extracellular Na+ concentrations, and consecutively a decrease of NALCN depolarizing function as we have previously demonstrated in this cell line [36].“
"Figure 1c, why does the injection of HBSS alone produce a tiny elevation of Ca around the second 40?"
Answer : We have already seen this weak response of HBSS/ buffer injection. It is not block by any drugs (data not shown). We believe that it is due to a mechanical effect (bulk effect?) of the injection on the cells.
Minor points:
- The real VGCC agonist is the levo enantiomer of Bay K, i.e,. (S)-(-)-BAY K8644. The dextro enantiomer is indeed a VGCC blocker. For economic issues, it is widely sold as the racemic mixture because the agonist effect of the S-enantiomer is more potent than the blocking effect of the R-enantiomer.
Answer : We agree, we precised this in 2.1 section.
- chapter 2.3.: Are authors sure that concentration of both antibiotics is 1 mM?. It seems too much.
Answer : It is an error. We added the correct concentrations : 1% penicillin/streptomycin
- TYPOS:
- Line 37: regualtion : Corrected
- Line 187: depolarizatoin: Corrected

Reviewer 2 Report
After reading the manuscript carefully, I concluded that the presented discussion requires a key explanation of the discussed measures of structural similarity of the studied / discussed compounds.
I strongly recommend the authors to perform a simple similarity test by calculating the Tanimoto Structural Similarity Index (Tsc) using the online software https://chemminetools.ucr.edu/similarity/. I am asking for a tabular summary of the obtained data and their substantive discussion.
While for pairs, e.g. liriodenine and oxostefanine, Tsc = 0.73; lauterine and oxostefanine Tsc = 0.83 we can talk about structural similarity (line no. 357). For the analyzed pairs cefarantin and tetrandrin Tsc = 0.13, tetrandrine and thaliphylline Tsc = 0.03 (line no. 368), we cannot talk about such similarity.
Additionally, the small number of citations of works from the last 5 years is quite surprising and such a situation is unacceptable in contemporary scientific research / documents, especially in the research papers. This manuscript summarizes many old research papers (out of 51 references, only 4 were published after 2017, which is only 8% of all sources cited). Therefore, I am convinced that the authors must carry out a detailed literature review and supplement the citations with references to the latest research in this field.
After clarifying these points and appropriate manuscript corrections, the work may be reconsidered for publication.
I recommend the publication after major corrections.
Author Response
Dear Reviewer 2,
We would like to thank you for your peer-reviewing, which helped us to improve this manuscript.
After reading the manuscript carefully, I concluded that the presented discussion requires a key explanation of the discussed measures of structural similarity of the studied / discussed compounds. I strongly recommend the authors to perform a simple similarity test by calculating the Tanimoto Structural Similarity Index (Tsc) using the online software https://chemminetools.ucr.edu/similarity/. I am asking for a tabular summary of the obtained data and their substantive discussion.
While for pairs, e.g. liriodenine and oxostefanine, Tsc = 0.73; lauterine and oxostefanine Tsc = 0.83 we can talk about structural similarity (line no. 357). For the analyzed pairs cefarantin and tetrandrin Tsc = 0.13, tetrandrine and thaliphylline Tsc = 0.03 (line no. 368), we cannot talk about such similarity.
Answer : as requested, such analysis were performed. We added a new figure showing the multidimension scaling analysis of structural similarity between selected IA and two tables illustrating Atom pair Tanimoto coefficients. A new paragraph was added to described these data. The discussion was then accordingly modified.
Additionally, the small number of citations of works from the last 5 years is quite surprising and such a situation is unacceptable in contemporary scientific research / documents, especially in the research papers. This manuscript summarizes many old research papers (out of 51 references, only 4 were published after 2017, which is only 8% of all sources cited). Therefore, I am convinced that the authors must carry out a detailed literature review and supplement the citations with references to the latest research in this field.
Answer : As requested, we swapped some old references with more recent ones.
Reference 2 was replaced by Catterall et al., 2020
Reference 6 was replaced by Qian et al, 2002
Réferences 10 and 11 were replaced by Bhagya et al., 2016 doi:10.1016/j.phytochem.2016.02.005
Reference 22 was replaced by Zhu et al, 2020
References 34 (2002) and 35 (2008) were exchanged by Vetter 2012, Basset 2017.
References 47 (Hussain et al, 1982) was removed, since it is not necessary.
We would like to keep the references cited in Table 1, since they correspond to the original papers that described the purification and structure in the litterature.
Round 2
Reviewer 2 Report
I read very carefully the authors' responses. I also read through the new version of the article. I believe that the authors addressed all the comments in a substantive and exhaustive manner, placing appropriate corrections in the body of the article. Therefore, I recommend this article for publication.